# CXCR4-CXCL12-CXCR7 and PD-1/PD-L1 in Pancreatic Cancer: CXCL12 Predicts Survival of Radically Resected Patients

**DOI:** 10.3390/cells11213340

**Published:** 2022-10-22

**Authors:** Crescenzo D’Alterio, Alessandro Giardino, Giosuè Scognamiglio, Giovanni Butturini, Luigi Portella, Giuseppe Guardascione, Isabella Frigerio, Marco Montella, Stefano Gobbo, Guido Martignoni, Vincenzo Napolitano, Ferdinando De Vita, Fabiana Tatangelo, Renato Franco, Stefania Scala

**Affiliations:** 1Microenvironment Molecular Targets, Istituto Nazionale Tumori IRCCS “Fondazione G. Pascale”, 80131 Naples, Italy; 2Unit of HPB Surgery, Pederzoli Hospital, Peschiera del Garda, 37019 Verona, Italy; 3Pathology Istituto Nazionale Tumori IRCCS “Fondazione G. Pascale”, 80131 Naples, Italy; 4Pathology Unit, Department of Mental and Physical Health and Preventive Medicine, University of Campania “Luigi Vanvitelli”, 80138 Naples, Italy; 5Department of Pathology, Pederzoli Hospital, Peschiera del Garda, 37019 Verona, Italy; 6Department of Translational Medicine, University of Ferrara, 44121 Ferrara, Italy; 7Department of Translational Medical Sciences, University of Campania “Luigi Vanvitelli”, 80138 Naples, Italy; 8Medical Oncology, Department of Precision Medicine, University of Campania “Luigi Vanvitelli”, 80138 Naples, Italy

**Keywords:** pancreatic adenocarcinoma (PDAC), CXCR4-CXCL12-CXCR7 axis, immune checkpoint PD-1/PD-L1, chemokine receptor, tumor microenvironment (TME), outcome, pancreatic resection, pancreatic cancer prognosis

## Abstract

Pancreatic ductal adenocarcinoma (PDAC) is currently the most deadly cancer. Although characterized by 5–20% of neoplastic cells in the highly fibrotic stroma, immunotherapy is not a valid option in PDAC treatment. As CXCR4-CXCL12 regulates tumor invasion and T-cell access and PD-1/PD-L1 controls immune tolerance, 76 PDACs were evaluated for CXCR4-CXCL12-CXCR7 and PD-1/PD-L1 in the epithelial and stromal component. Neoplastic CXCR4 and CXCL12 discriminated PDACs for recurrence-free survival (RFS), while CXCL12 and CXCR7 discriminated patients for cancer-specific survival (CSS). Interestingly, among patients with radical resection (R0), high tumor CXCR4 clustered patients with worse RFS, high CXCL12 identified poor prognostic patients for both RFS and CSS, while stromal lymphocytic-monocytic PD-L1 associated with improved RFS and CSS. PD-1 was only sporadically expressed (<1%) in focal lymphocyte infiltrate and does not impact prognosis. In multivariate analysis, tumoral CXCL12, perineural invasion, and AJCC lymph node status were independent prognostic factors for RFS; tumoral CXCL12, AJCC Stage, and vascular invasion were independent prognostic factors for CSS. CXCL12’s poor prognostic meaning was confirmed in an additional perspective-independent 13 fine-needle aspiration cytology advanced stage-PDACs. Thus, CXCR4-CXCL12 evaluation in PDAC identifies prognostic categories and could orient therapeutic approaches.

## 1. Introduction

Pancreatic ductal adenocarcinoma (PDAC) represents the fourth leading cause of cancer-related deaths in economically advanced countries [1]. In the European Union (EU), PDAC has increased by 5% between 1990 and 2016, the highest increase in the EU’s top five cancer killers [2,3]. The minority of PDACs (20%) are resectable, 30% present with locally advanced disease, and 50% are metastatic at diagnosis [4]. The overall 5-year survival is 10% and, among the small subset of radically resected patients (R0), is about 20% [5]. Tumor size, radical surgery, histological differentiation, vascular–perineural invasion, and lymph node involvement [6,7] are evaluated as prognostic factors. The low rate of long survival after pancreatic resection and the high percentage of early progressor in resectable stages demand deeper insights into the PDAC biology. A molecular classification identifies four major driver genes implicated in PDAC tumorigenesis: KRAS somatic mutations, mostly clustered in codon 12, identified in more than 90% of PDAC [8], CDKN2A (the most frequently mutated tumor-suppressor gene, with loss of function in over 90%) and TP53 [8]. SMAD4, promoting tumor-suppressor effects of the TGFβ receptor, is inactive in 50% of the PDAC [8]. Nevertheless, KRAS, TP53, SMAD4 and CDKN2A mutations do not identify subtype-specific treatment options [9]. PDACs are non-inflamed tumors with low immunogenicity, also known as “cold tumors”, in which T cells are excluded [10,11]. Apart from the mismatch repair deficient subtype (MMR-D), PDACs show a median mutational burden of 4, standing for low neoantigen presentation [11]. Histologically PDAC often contains only 5–20% of neoplastic cells intermixed with dozens of stromal cell types [12] and fibrotic stroma that can physically confine cytotoxic T cells away from tumor cells. Moreover, the immunosuppressive microenvironment within the stroma can suppress infiltrating T cells activity [13,14], rendering immunotherapy limited in PDAC [15,16]. The value of PD-L1 remained inconsistent in PDAC either as prognostic and/or predictive of immunotherapy response [17,18,19]. Cancer-activated fibroblasts (CAFs)-released CXCL12 chemokine activates CXCR4 and/or CXCR7 receptors and regulates PDAC cell proliferation, migration, and invasion [20,21,22,23,24,25]. Pooled studies on the prognostic role of CXCL12 [17,26,27,28] recognized CXCL12 as a poor prognostic factor in PDAC [29]. Califano recently identified three clearly distinct cellular subtypes, morphogenic, lineage, and oncogenic precursors, in 200 human primary PDACs [23]. Morphogenic cells lost gastrointestinal (GI) epithelial features and gained undifferentiated mesenchymal characteristics overexpressing CXCR4, SNAI1, ZEB1, and ZEB2. By contrast, oncogenic precursors are more differentiated than lineage and morphogenic, exhibiting low stemness markers MSI2, PROM1/CD133, and CXCR4 expression [23]. The aim of the study is to evaluate the possible prognostic value of cancer and stromal CXCR4-CXCL12-CXCR7 axis and PD-1/PD-L1 in 76 consecutive patients undergoing upfront resection for pancreatic carcinoma.

## 2. Materials and Methods

### 2.1. Patients and Tissues

Patients with resectable PDAC [30] consecutively undergone upfront surgery from January 2014 to April 2015 were retrospectively reviewed. Seventy-six pancreatectomies were obtained at Ospedale P. Pederzoli Peschiera del Garda (Verona), Italy. A total of 56 patients (73.7%) underwent pylorus-preserving pancreaticoduodenectomy (PP-PD), 6 patients (7.9%) underwent classical Whipple pancreaticoduodenectomy (PD), 14 patients (18.4%) underwent to distal pancreatectomy and splenectomy (DP). Complete tumor resection, defined as the absence of tumor cells within 1 mm from the surgical margin, was achieved in 51(67.1%) patients (R0), while the presence of tumor cells within 1 mm of the resection margin (R1) in 25 (32.9%) patients.

Three-micrometer sections were cut from formalin-fixed paraffin-embedded (FFPE) tissue blocks. Sections were stained with hematoxylin and eosin for adequate tumor representation.

### 2.2. Immunohistochemistry

FFPE sections were dewaxed and rehydrated, and heat-induced epitope retrieval (HIER) (Decloaking Chamber™ NxGen Biocare Medicals) with appropriate Antigen Unmasking Solution was performed. After incubation with the appropriate serum for blocking non-specific background, FFPE tumor tissue slides were incubated overnight at 4 °C using primary antibodies: Mouse Monoclonal anti-human Anti-CXCR7/RDC-1 Antibody ((11G8), HIER citrate buffer pH6, 1:50 dilution, R&D Systems); Mouse Monoclonal anti-human Anti-CXCL12/SDF-1 Antibody ((79018), HIER citrate buffer pH6, 1:50 dilution, #MAB350 R&D Systems); two commercial CXCR4 antibody clones the mouse Monoclonal anti-human Anti-CXCR4/CD184 ((44716) HIER, citrate buffer pH6, 1:100 dilution, #MAB172 R&D Systems) and the rabbit monoclonal anti-human Anti-CXCR4 ((UMB-2) HIER, Tris-EDTA buffer, pH 9.0, 1:200 dilution, Abcam); two commercial PD-L1 antibodies; the rabbit monoclonal anti-human anti-PD-L1/CD274 ((E1L3N) 1:200; diluted; HIER citrate buffer pH6, #13684 Cell Signaling Technology and (22C3) ready to use with HIER Conditioning Solution (CC1) pH9 Ventana Medical Systems); Mouse monoclonal anti-human PD-1/CD279 ((NAT-105) ready to use, HIER Cell Conditioning Solution (CC1) pH9, Ventana Medical Systems). Stained pancreatic cancer cells were assessed in at least three regions of interest (ROI)/slide, recognized at low power (100× magnification), and the cells enumerated in 5 consecutive, not-overlapping high-power fields (400× magnification—0.237 mm^2^/field) for each ROI, on Olympus BX51 microscope (Olympus, Tokyo, Japan). The evaluation was carried out by 3 qualified observers (FT, CD, and MM). For CXCR7 and CXCL12, staining extension was calculated for each case by the average percentage of positively stained cancer cells for the three ROIs. For CXCR4, due to heterogeneous staining patterns, the staining value was calculated by H-score. Staining intensity was based on membrane staining and rated as absent (0), weak/low (1+), and intermediate/moderate (2+), typically with cytoplasmic localization, strong/high (3+), typically cytoplasmic with membrane localization of CXCR4 staining. The % of positive cells at each staining intensity is obtained, and an H-score is determined by the sum of each intensity rating multiplied by its corresponding percentage. CXCR4 was scored as positive at H-score > 50, corresponding to PDAC with multiple subcellular signal accumulation (membrane and cytoplasm). CXCR7 and CXCL12 cells were rated positive when stained regardless of the cellular localization. CXCR7 was scored as positive with >20% positive cells [31]; CXCL12 was scored positive with >5% positive cells. PD-L1 expression in tumor cells was evaluated and scored as positive if membrane staining > 5%. Stromal cells were rated positive if specifically stained and scored as positive if >5%.

### 2.3. Statistical Analysis

Association between CXCR4, CXCL12, CXCR7, and PD-1/PD-L1 expression cancer cells and patients’ clinic-pathological features were analyzed applying chi-square and Mann–Whitney U tests for categorical and continuous variables, respectively. Recurrence-free survival (RFS) was set as the time from diagnosis to the recurrence or last follow-up. Cancer-specific survival (CSS) was set as the time from diagnosis to death for cancer. Kaplan–Meier method was performed to estimate the survival curve and logrank test for statistical comparison. Cox proportional hazards regression was utilized to test the effect of multiple dichotomous covariates (risk factors) on RFS and CSS; The backward method for variable selection was applied in the final model with a conventional *p*-value threshold of *p* < 0.05 (enter variable), and *p* > 0.1 (remove variable). *p*-values less than 0.05 were considered significant. All statistical tests and graphs were conducted using SPSS version 20 (IBM Corp., Armonk, NY, USA) and MedCalc 12.3.0 (MedCalc. Software; Mariakerke, Belgium).

### 2.4. Validation Cohort and Tissues

A population of twenty patients diagnosed with advanced PDAC, evaluated through endoscopic-US-guided pancreatic fine-needle aspiration cytology (FNAC), was considered a validation cohort. The samples were collected and analyzed at Pathology Unit, Department of Mental and Physical Health and Preventive Medicine, University of Campania “Luigi Vanvitelli”, Naples, Italy, from January 2020 to October 2021. Three-micrometer sections were stained for CXCL12 immunocytochemical evaluation following hematoxylin/eosin reviewing and histological confirmation. All cases were retrospectively reviewed, and the cytology on direct smears showed moderately to richly cellular samples composed of three-dimensional aggregates of epithelial cells with severe cytological atypia. The cyto-block (CB) sections were also re-examined, with a cytological morphology substantially similar to that observed on direct smears; when necessary, immunocytochemical (ICC) analysis was performed, which showed positivity in the neoplastic cells for CK19 and CA 19–9. In all 20 cases selected, the overall morphological analysis was consistent with the diagnosis of pancreatic adenocarcinoma (Category VI sec. Papanicolaou System). ImmunoCytoChemistry (ICC) was carried out on CB sections and stained for CXCL12 as previously described.

## 3. Results

### 3.1. Epithelial and Stromal PDAC Cells Express CXCR4

Clinical pathological characteristics of the 76 patients are presented in Table 1.

CXCR4 was highly expressed in 25 out of 76 tumors (32.9%) (Table 2). CXCR4 membrane and/or cytosolic staining was evaluated, and CXCR4 staining was reported as H-score (% positive cells X staining intensity). The mean CXCR4 expression was 46.88 ± 51.23 H-score units (median 51.23, range 5–210) (Table 2).

Representative low–moderate and high CXCR4 expressions were reported (Figure 1C–H). CXCR4 expression was reported in 53/76 (69.7%) of tumor-infiltrating cells (Table 2). Extensive dense fibrotic stroma/desmoplasia or perineural invasion associated with PDAC did not express CXCR4 (Figure 1I,J). CXCR4 staining was reported in acinar cells belonging to the exocrine pancreas (Figure 1K,L) and islets of Langerhans surrounded by dense fibrotic stroma or desmoplasia (reported in 47.3% of PDAC TME) (Figure 1M,N). CXCR4 was also detected on lymphocytic monocytes and endothelial infiltration in 53/76 tumors (69.7%) with CXCR4 (Figure 1O,P).

### 3.2. Epithelial and Stromal PDAC Cell Express CXCR7

CXCR7 was highly expressed in 26 out of 76 (34.2%) tumors and lowly expressed or undetectable in 50 tumors (65.8%). The CXCR7 mean expression was 13.3 ± 16.7% (Table 2). Unlike CXCR4, CXCR7 the expression was mainly cytosolic (Figure 2A–H). More than 50% of the desmoplastic stromal cells were positive for CXCR7. CXCR7 expression was observed in neural cells with neoplastic invasion (Figure 2K,L), in desmoplasia-trapped islets of Langerhans cells, reported in 33/76 (43.3%) of PDAC TME, in lympho-monocytic/endothelial infiltration, reported in 43/76 (56.6%) (Figure 2M,N) and in acinar cells belonging to the exocrine pancreas (Figure 2O,P).

### 3.3. PDAC Cancer Cells Express CXCL12

CXCL12 was predominantly identified in the membrane in 20 out of 76 (26.3%) tumors. The CXCL12 mean expression was 3.7 ± 7.20% (Table 2). Representative CXCL12 expression is reported in Figure 3A–H). Tumor-infiltrating stroma cells were CXCL12 positive in only 15/76 (19.7%) (Figure 3I–N). Robust CXCL12 expression was observed in exocrine pancreatic ductal cells adjacent to PDAC (Figure 3J).

### 3.4. PD-L1 Is Predominantly Expressed by TME in PDAC

PD-1 and its natural ligand PD-L1 were evaluated in 76 PDAC. While PD-1 was not detectable in cancer cells, it was only sporadically expressed (<1%) in focal lymphocyte infiltrate (data not shown). PD-L1 was predominantly identified at the membrane of cancer cells in 29/76 (38.2%) PDAC. PD-L1 mean expression was 3.4 ± 7.1 (Table 2). (Figure 4C–H). Tumor-infiltrating stroma cells were PD-L1 positive. TME and acinar normal cells surrounding cancer cells highly expressed PD-L1 (Figure 4I–J). Intense homogeneous and consistent PD-L1 staining was detected in the nerve bundle (Figure 4K,L). PD-L1 staining was observed in infiltrating inflammatory cells in 32/76 (42.2%) PDAC (Figure 4M–P).

### 3.5. Prognostic Significance of Epithelial and Stroma CXCL12, CXCR4, CXCR7 and PD-L1 in PDAC

The CXCL12 and CXCR4 tumor expression positively correlated (Spearman’s *r* = 0.42, *p* = 0.004). The relation among the evaluated markers and clinic pathologic characteristics revealed that CXCR4-positive tumor and CXCR4-positive tumor-infiltrating inflammatory cells were associated with vascular invasion (*p* = 0.0421 and 0.045) as expected [32] (Table 3). Tumor-infiltrating inflammatory cells CXCR7 positive were mainly detected in smokers (*p* = 0.002) (Table 3), as previously reported [33].

Median follow-up was 25.02 months for recurrence-free survival (RFS) (95% confidence interval (CI): 20.19–29.85); 33.60 months for cancer-specific survival CSS (95% confidence interval (CI): 28.907–38.998). A total of 71 PDAC were analyzed for RFS; 58/71 (81.7%) patients experienced recurrence, with mean RFS durations of 17.30 months, and 13 (18.3%) did not experience recurrence up to 60.52 months. Out of 76, 4 patients were excluded for cancer-unrelated death; 72 PDAC were analyzed for CSS, 56/72 (77.8%) with mean CSS durations of 25.30 months, and 16/72 (22.2%) patients were alive at a mean of 62.65 months. Univariate analyses of RFS and CSS are summarized in Table 4.

Patients with high CXCR4 expression had a significantly worse outcome (RFS: 11.76 months vs. 26.0 months *p* = 0.0024; CSS: 21.9 months vs. 36.6 months *p* = 0.08, not significant) (Figure 5A).

Of note, high CXCR4 predicted short RFS in R0 subgroup (radical resected, absence of neoplastic cells within 1 mm from the lesion) (*n* = 48) (RFS: 11.05 months vs. 33.70 months *p* = 0.0071) (Figure 6A). CXCL12 expression predicted RFS (positive CXCL12: 10.0 months vs. negative CXCL12: 33.7 months; *p* = 0.0001) and CSS (positive CXCL12: 19.2 months vs. negative CXCL12: 35 months; *p* = 0.0024) (Figure 5B). Moreover, CXCL12 predicted short RFS (*p* = 0.0001) (*n* = 48, PFS data were unavailable for 3/51 R0 patients) and CSS (*p* = 0.017) (*n* = 50, as 1 out of 51 R0 died from other cause than pancreatic cancer) in R0 patient’s subgroup (Figure 6B). Patients with high CXCR7 expression had a short RFS (*p* = 0.0441) but not CSS (*p* = 0.0605) (Figure 5C). PD-L1 did not show prognostic potential (Figure 5D).

While the expression of the CXCR4-CXCL12-CXCR7 axis or PD-L1 in stromal cells was not prognostic in the whole cohort (Appendix A), we found a significant association of high stromal lymphocytic-monocytic PD-L1 expression with improved RFS (*p* = 0.016) and CSS (0.047) in R0 patients, (Figure 7).

### 3.6. Tumoral CXCL12 Is an Independent Poor Prognostic Factor at the Multivariate Analysis

In multivariate analysis (Table 5), the pathological AJCC Stage 8th ed. (*p* = 0.049), perineural invasion (*p* = 0.0107), and CXCL12 (*p* = 0.00002) but not CXCR4, predicted shorter poor RFS (Table 5 left). The pathological AJCC Stage 8th ed. (*p* = 0.00007), vascular invasion (*p* = 0.0067), and CXCL12 (*p*= 0.0062) but not CXCR7 nor CXCR4 expression predicted shorter poor CSS (Table 5 right); thus, CXCR7 or CXCR4 were not retained in the final multivariate models of RFS and CSS.

To confirm the CXCL12 prognostic role, CXCL12 expression was evaluated in an independent cohort of FNAC (*n* = 20). The median age was 72 ± 10 years (range 50–85), with 8 (40%) male and 12 (60%) female, and 17 (85%) of patients ≥60 years old. At diagnosis, the majority of patients (85%) had an advanced stage. CXCL12 was detected in 70% (14/20) PDACs predominantly at the membrane and cytoplasm of cancer cells (Figure 8) and in some surrounding monocytes/macrophages stromal cells. A subgroup of 8 out of 20 (40.0%) showed a consistent membranous cytoplasmic distribution of CXCL12 (mean expression 86.9 ± 11.9%) (Figure 8A,B). CXCL12 was negative/low in 12 out of 20 (60%); absent in 6 out of 20 FNAC PDAC (30.0%) and low 6 out of 20 showed low broad distribution (36.66 ± 12.11%) (Figure 8C,D). High CXCL12 was also detected in tumor emboli (Figure 8E) and perineural infiltration (Figure 8G).

Follow-up was available for 13 patients. A total of 7 patients died (53.8%), showing a median overall survival of 6 months. The patients with a high sharp, consistent distribution of CXCL12 expression showed short CSS with a median survival of 3 months, while low/negative CXCL12 expressing displayed short CSS with a median survival of 12 months (*p* = 0.029) (Figure 8).

## 4. Discussion

The role of the CXCR4-CXCL12-CXCR7 axis and PD-1/PDL-1 was addressed in tumor/stromal cells in 76 consecutive single-center patients undergone surgery between January 2014/March 2015 and followed for 5 years until January 2021. We demonstrated that the entire axis CXCR4-CXCL12-CXCR7 was overexpressed in PDAC neoplastic cells as compared to TME cells. Moreover, CXCL12 significantly correlated with poor prognosis in an unrelated cohort of 20 FNAC from PDAC patients. CXCR4 was expressed by acinar cells and islets of Langerhans, surrounded by extensive dense fibrotic stroma or desmoplasia [34]. Trefoil Factor 2, an exocrine gene early expressed during mouse embryonic development, induced CXCR4 activation in acinar cells promoting cell proliferation and preventing apoptosis [34]. CXCR4 was also detected in lymphocytic-monocytic infiltration and endothelial cells [35] associated with vascular invasion [36]. Thus, it is possible to hypothesize that CXCR4 imaging might detect inflammatory and vascular involvement in PDAC and surgical/clinical settings [37,38]. CXCR7 is upregulated in tumor and stromal PDAC. Schwann cells, CXCR7 positive, were reported in pancreatic nerves and CXCR7-neutralizing antibody (9C4), but not CXCR4 inhibitor AMD3100, induced significant dose-dependent attenuation of transmigration toward PDAC cell lines [39]. Herein, CXCR7 expression was reported in nerve bundle, and CXCR7 expression of nerve bundle, in inflamed endothelial cells [40]. Of note, CXCR7 was significantly more expressed in the PDAC stroma of patients with a smoking habit as nicotine induces ERK-dependent IL-8-activating stromal nicotinic receptors [33,41] and IL-8 induces CXCR7 expression [42,43]. CXCL12 expression was observed mainly in tumor cells and, in some cases, in cancer-associated fibroblasts. Hypoxia induces CXCL12-CAF release and recruitment of CXCR4-positive immune-suppressive/supportive stromal cells [39,44]. Chronic pancreatitis was previously reported to be a risk factor for PDAC [45,46]. Long-standing inflammation increases cell turnover and stellate cell proliferation, promoting carcinogenesis [47,48]. Nevertheless, it was not prognostic in our cohort (*p* ≥ 0.05) as for NCCN 2022 guidelines. In our analysis, chronic pancreatitis was a “potential risk factor” (univariate analyses *p*-value 0.0365) not retained, for irrelevance or redundancy, in the multivariate analysis. As for PD-L1, early studies showed prognostic significance for overall survival with >5 or 10% cut-off [49,50]. Although evidence suggests that PD-L1 is an indicator of poor prognosis, different patient selections and variable thresholds (1–10%) do not allow comparison among the reports. Basile Tessier-Cloutier reported three values for PD-L1 tumor cell membrane staining (≥1%, >5% and >10%) on tissue microarray evaluating cancer-specific survival on 252 PDACs with 12 PDACs-PD-L1 overexpressing (>10%) displaying poor survival [51].

Karamitopoulou reported PD-L1 expression in about one-third of 349 samples. The authors concomitantly analyzed PD-L1, PD-1, CD3, CD4, CD8, FoxP3, and CD68, reporting that PD-L1, present with T-cell infiltration, improved overall survival [52]. Herein, a similar pattern, although not significant, was reported for RFS and OS. The possible discrepancy may be ascribed to patients’ subgroup analysis (mismatch repair (*MMR*) *proteins*) and sample size. In agreement with the herein reported data, Diana et al. examined the prognostic value of PD-1 and PD-L1, together with CD8+ tumor-infiltrating lymphocytes (TILs) and FOXP3+ Tregs in 145 PDAC samples, describing PD-L1 expression not prognostic [53]. Wang et al. reported that PD-L1 expression failed to predict prognosis in 77 unselected PDACs, although a further subgroup analysis (high/low PD-L1 and tumor density) identified patients with worse overall survival [54]. As related to PD-1, rare PD-1-positive isolated T cells were revealed as expected by the low mutation burden in PDAC. Thus, we did not pursue more combined staining.

PD-L1 was recently reported in stromal cells in PDAC [55], and also, here, PD-L1 was reported on nerve bundle of desmoplastic stroma and affected the prognosis in a subgroup of R0 patients. Although previous studies reported a prognostic role of tumor PD-L1 expression, we could not observe it [55]. PDAC tumor microenvironment (TME) plays a central role in impairing the efficacy of immune checkpoint inhibition (ICI) [56]. Adding CXCR4 blockade to PD-1 targeted therapy increased tumor cell death concomitantly with lymphocyte expansion [56], thus ongoing trials are evaluating the efficacy of targeting CXCR4–CXCL12 axis in potentiating the immune checkpoint inhibitors-based therapy in pancreatic cancer (NCT03277209, NCT02907099 NCT04177810). This manuscript has several limitations. The retrospective nature and the patients number do not allow definitive conclusions. Although the patients were consecutively enrolled, they represent a selected group of patients for staging and fitness for surgery. A comparison with unresectable patients could represent the natural prosecution of the study. Nevertheless, this is the first time that the CXCR4-CXCL12-CXCR7 axis discriminates patient outcomes while PD-L1 expression both in cancer or stromal cells did not have significant predictive ability for survival; further, PDACs did not express PD-1. CXCL12 expression was an independent prognostic factor for RFS and CSS. Moreover, CXCL12 in an unrelated cohort of 20 FNAC from PDAC patients confirmed a significant correlation with poor prognosis. CXCL12 expression was an independent prognostic factor for RFS and CSS. Moreover, CXCL12 in an unrelated cohort of 20 FNAC from PDAC patients confirmed a significant correlation with poor prognosis. Notably, high CXCR4 and high CXCL12 identified high-risk patients among the radically resected patients, regarded as having good prognosis; thus, CXCR4 CXCL12 evaluation could be useful for R0 patient risk assessment and potentially allow to discriminate after a prospective validation of our results, those who could potentially better benefit from neoadjuvant chemotherapy among resectable stages and, conversely, identify those who can better benefit from upfront surgery. Herein high stromal lymphocytic-monocytic PD-L1 expression is associated with improved RFS in the R0 subgroup. It is hypothesized that the exhausted immune population is still exerting control over tumor growth, as recently reported [52,57], suggesting a potential benefit for PD-1/PD-L1 antagonists in this subgroup. To the best of our knowledge, this is the first study that systematically investigates on tumor vs. stromal prognostic role of CXCR4-CXCL12-CXCR7 and PD-L1/PD-1 in PDAC. Targeting the axis CXCR4-CXCL12-CXCR7 thus represents a suitable tool to improve diagnosis and obtain prognostic and/or predictive data for optimizing combined therapy in PDAC patients.

## Figures and Tables

**Figure 1 cells-11-03340-f001:**
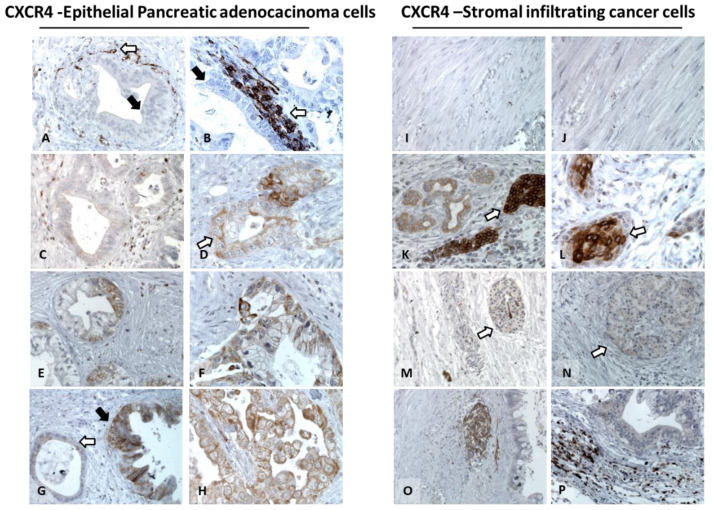
CXCR4 expression in epithelial and stromal PDAC cells. Representative expression in PDAC (200× and 400× magnification). Left panel: CXCR4-negative PDAC (**A**) and positive lympho-monocytic cells (white arrow) (**B**). CXCR4 low, cytoplasmic, and few membrane stainings (white arrow) (**C**,**D**). CXCR4 moderate expression, heterogeneous mild diffusion, and intensity of membrane and cytoplasmic (**E**,**F**). CXCR4 high expression, featured by a diffuse and strong membrane, cytoplasmic staining (black arrow), compared to the adjacent lesion with a low degree of cytological and architectural degeneration and low CXCR4 expression (white arrow) (**G**,**H**). TME analysis (right panel): Fibrotic stroma negative (**I**,**J**), CXCR4 expression in acinar cells (white arrow) (**K**,**M**). Islets of Langerhans (white arrow) (**M,N**) with faint CXCR4 surrounded by negative fibrotic stroma (white arrow), clear CXCR4 expression in lymphocytic monocytes, and endothelial infiltration (**O**,**P**).

**Figure 2 cells-11-03340-f002:**
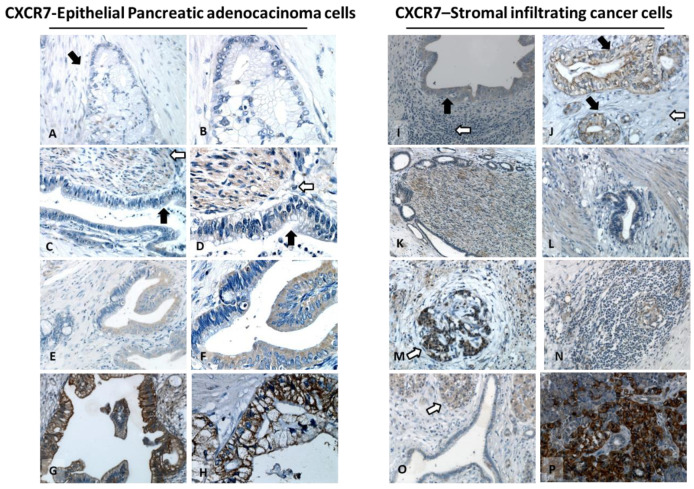
Epithelial and stromal PDAC cells express CXCR7**.** Representative expression in PDAC (left panel 200x and 400x magnification): CXCR7-negative sample (black arrow) (**A**,**B**). CXCR7 low expression, with cytoplasmic staining with low diffusion (black arrow) (**C**,**D**), adjacent to CXCR7-positive nerves (white arrow). CXCR7 moderate expression, featured by heterogeneous mild diffusion, cytoplasmic CXCR7 staining (**E**,**F**). CXCR7 high expression, with diffuse and strong membrane and cytoplasmic staining (**G**,**H**). TME analysis (right panel): CXCR7-negative sample (white arrow), with positive PDAC (black arrow) (**I**,**J**). CXCR7 expression in nerves adjacent to PDAC perineural invasion (50× magnification) (**K**) and in the fibrotic stroma (**L**) (200× magnification). CXCR7 expression was reported in desmoplasia-trapped islets of Langerhans cells (white arrow) (200× magnification) (**M**) in lympho-monocytic/endothelial infiltration (**N**) (200× magnification) and in acinar cells belonging to the exocrine pancreas (white arrow) (**O**,**P**) (200× and 400× magnification).

**Figure 3 cells-11-03340-f003:**
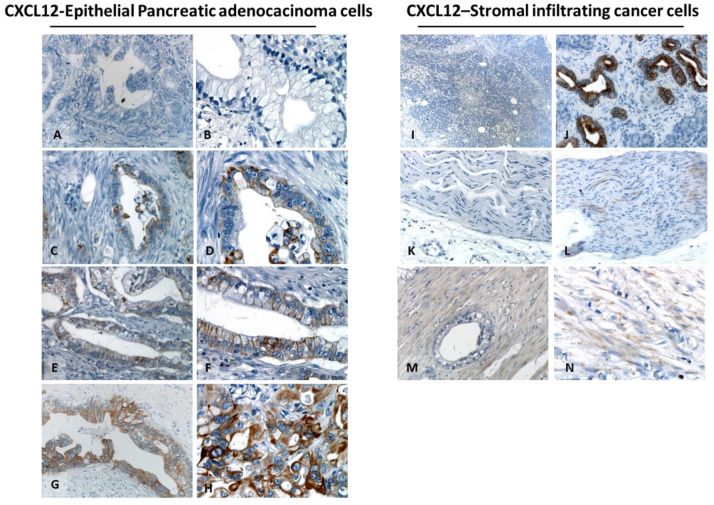
PDAC Cancer cells express CXCL12. Representative expression in PDAC (left panel 200× and 400× magnification). CXCL12 negative (**A**,**B**). CXCL12 low expression, featured mainly by membrane and cytoplasmic staining (**C**,**D**). CXCL12 moderate expression, featured by mild diffusion of membrane and cytoplasmic staining (**E**,**F**). CXCL12 high expression, featured by massive diffusion membrane and cytoplasmic staining (**G**,**H**). TME analysis (right panel): Representative CXCL12-negative sample (50×) (**I**), with strong positive signal in TME exocrine pancreatic ductal cells adjacent PDAC (100× magnification) (**J**). No (**K**) or faint (**L**) staining in the nerve (200× magnification), CXCL12 expression was reported in fibrotic tissue (M,**N**).

**Figure 4 cells-11-03340-f004:**
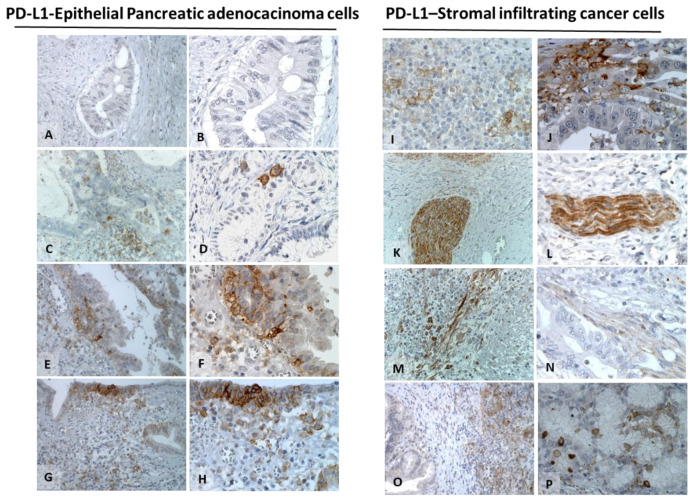
PDL-1 is predominantly expressed by TME in PDACs. Representative expression in PDAC (left panel 200× and 400× magnification). PD-L1-negative sample (**A**,**B**). PD-L1 low expression, featured by membrane staining with low diffusion (**C**,**D**). PD-L1 moderate expression, featured by heterogeneous diffusion of membrane staining (**E**,**F**). PD-L1 high expression, featured by diffused and strong membrane staining in PDAC (**G**,**H**). TME analysis (right panel): Representative PD-L1-positive staining in lymphocytes and histiocytes surrounding PDACs (**I**,**J**). Intense homogeneous and consistent PD-L1 expression in nerves (**K**,**L**) (200× and 400× magnification) as a major source of PD-L1 in PDAC. PD-L1 was further reported in fibrotic tissue (**M**,**N**) (200× and 400× magnification) and in peritumoral lympho-monocytic infiltrating cells (**O**,**P**).

**Figure 5 cells-11-03340-f005:**
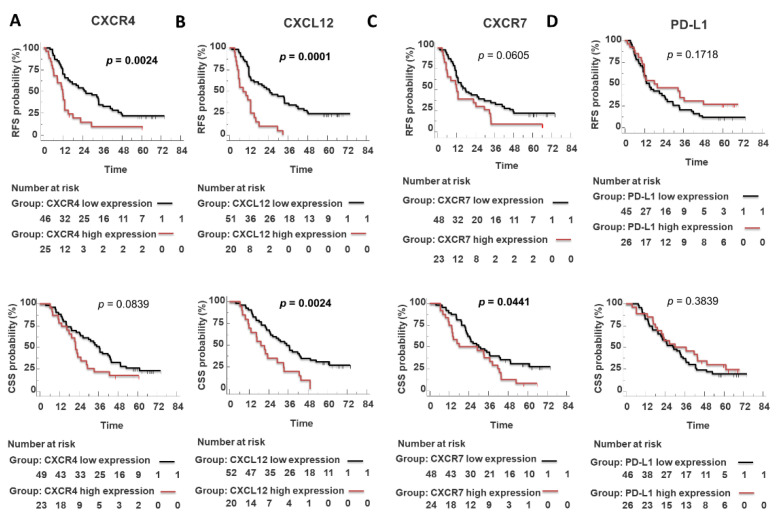
CXCR4, CXCL12, and CXCR7 expression but not PD-L1 predict poor survival in PDAC patients (**A**–**D**). Survival LogRank analysis correlates with recurrence-free survival (RFS) and cancer-specific survival (CSS) (months).

**Figure 6 cells-11-03340-f006:**
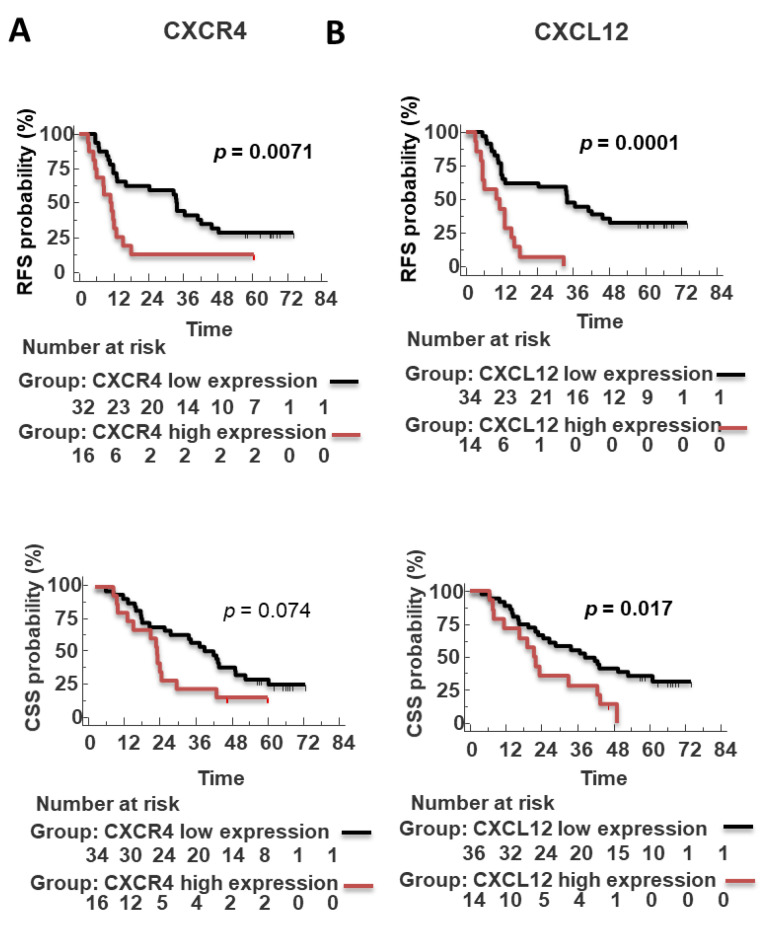
CXCR4 and CXCL12 in tumor cells predict poor prognosis in R0 patients. Survival LogRank analysis correlates recurrence-free survival (RFS) and cancer-specific survival (CSS) (months) of CXCL12 and CXCR4 (**A**,**B**) expression restricted to with radical resection (R0) PDAC patients subpopulation.

**Figure 7 cells-11-03340-f007:**
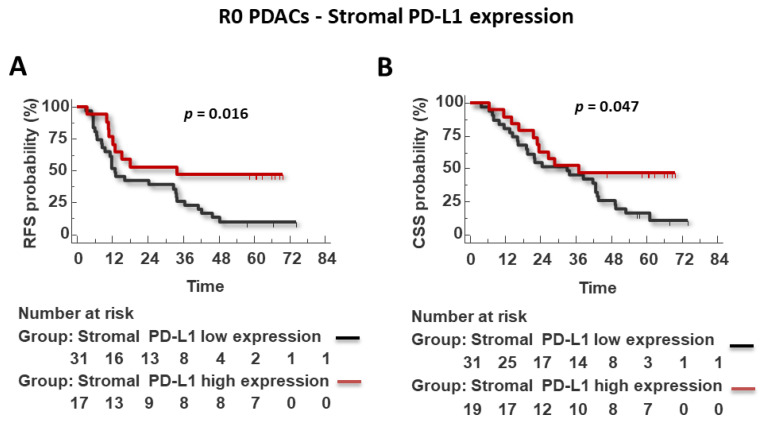
Stromal lympho-monocytic PD-L1 high expression predicts improved RFS and CSS survival in R0 patients. Survival LogRank analysis correlates with recurrence-free survival (RFS) (**A**) and cancer-specific survival (CSS) (months) (**B**).

**Figure 8 cells-11-03340-f008:**
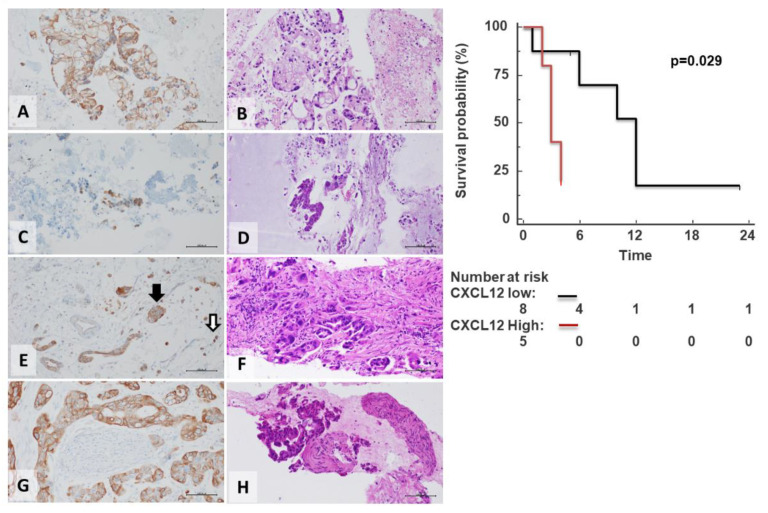
CXCL12 overexpression in FNAC-PDACs with poor prognosis. CXCL12 was predominantly identified in cancer cells. Representative microphotographs of membranous and cytoplasmic CXCL12 expression in PDAC FNAC (**A**,**C**,**E**,**G**) with related haematoxylin/eosin staining (**B**,**D**,**F**,**H**). CXCL12 high (**A**) and low cancer expression (**C**). Rare CXCL12-positive tumor-infiltrating was observed mainly surrounding monocytes/macrophages (white arrow), while robust cancer CXCL12 staining might be useful for visualizing tumor emboli (black arrow) (**E**) and PNI (**G**) (original magnification 200×).

**Table 1 cells-11-03340-t001:** Patients clinicopathological characteristics.

Variables	N of Patients (%)	N Patients
Age (yr); mean ± SD; median (range)	66.02 ± 9.6; 67 (42–84)	76
≤60	16(21.1%)	
>60	60(78.9%)	
Gender		76
Female	34(46.6%)	
Male	42(55.4%)	
Pancreas tumor location		76
Head	61 (80.3%)	
Others	15(19.7%)	
CA 19–9 (U/mL), mean ± SD; median (range)	337.8 ± 753.6; 88.6(0.6–4443)	76
0(0–100 U/mL)	39(51.3%)	
1(100–350 U/mL)	20(26.3%)	
2(>350 U/mL)	17(22.4%)	
Histologic Tumour grade		76
G2	69(90.8%)	
G3	7(9.2%)	
Histotype		76
Pancreatic ductal adenocarcinoma	72(94.7%)	
Pancreatic ductal adenocarcinoma + IPMN	4(5.3%)	
AJCC Stage 8th ed.		76
IA	6(7.9%)	
IB	6(7.9%)	
IIA	2(2.6%)	
IIB	28(36.8%)	
III	34(44.7%)	
Tumour size (mm), mean ± SD; median (range)	28.6 ± 9.8; 25(10–60)	76
0–20	19(25.0%)	
>20	57(75.0%)	
Positive lymph nodes	4.6 ± 4.9; 3(0–24)	76
N0(0)	14(18,4%)	
N1(1–3)	28(36.8%)	
N2(>4)	34(44.7%)	
R Status		76
R0	51(67.1%)	
R1	26(32.9%)	
Vascular neoplastic emboli		76
No	10(13.2%)	
Yes	66(86.8%)	
Vascular Invasion		76
No	65(82.9%)	
Yes	11(17.1%)	
Perineural Invasion		76
No	11(14.5%)	
Yes	65(85.5%)	
Adjuvant Chemotherapy		70
No	10(14.3%)	
Yes	60(85.7%)	
Adjuvant Radiotherapy		70
No	50(71.4%)	
Yes	20(28.6%)	

BMI: Body Mass Index (Kg/m^2^); IPMN: Intraductal papillary mucinous tumor (IPMN); American Joint Committee on Cancer (AJCC); R status, Surgical resection margin R0, tumor cells from margin distance > 1 mm, R1 tumor cells from margin distance ≥ 1 mm.

**Table 2 cells-11-03340-t002:** CXCR4, CXCR7, CXCL12 and PD-L1 expression in PDACs.

	CXCR4		CXCR7		CXCL12		PD-L1	
	Tumor	Stroma	Tumor	Stroma	Tumor	Stroma	Tumor	Stroma
N Patients assessable	76	76	76	76	76	76	76	76
mean (±SD)	46.88(±51.23)	13.4(±10.8)	13.3(±16.7)	3.6(±7.5)	3.7(±7.2)	1.2(±3.6)	3.4(±7.1)	5.8(±8.6)
Median (range)	37.5(0–210)	10(0–40)	5(0–50)	0(0–60)	0(0–30)	0 (0–20)	0(0–30)	0 (0–30)
Positive case (%)	25 (32.9%)	53(69.7%)	26(34.2%)	43(56.6%)	20(26.3%)	15(19.7%)	29(38.2%)	32(42.2%)
Negative case (%)	51(67.1%)	23(30,3%)	50(65.8%)	33(43.4%)	56(73.7%)	61 (80.3%)	47(62.8%)	44(57.9%)

**Table 3 cells-11-03340-t003:** CXCR4, CXCR7, CXCL12 and PD-L1 association with patient clinicopathological features.

	CXCR4		CXCR7		CXCL12		PD-L1	
	Tumor	Stroma	Tumor	Stroma	Tumor	Stroma	Tumor	Stroma
Age(yr) (≤60 vs. >60)	─	─	─	─	─	─	─	─
Tumor location (Head vs body/tail)	0.071	─	─	─	─	─	─	0.077
AJCC Stage 8th ed. (I–II vs. III)	─	─	─	─	─	─	─	─
Positive Lymph node (N0 vs. N1–2)	─	─	─	─	─	─	─	─
Smoke habit (yes vs. no)	─	─	─	0.002	─	─	─	─
Vacular invasion	0.042	0.045	─	─	─	─	─	─
Perineural invasion	─	0.056	─	─	─	─	─	─
American Joint Committee on Cancer (AJCC) Stage								

**Table 4 cells-11-03340-t004:** Univariate analysis of cancer specific survival and recurrence-free survival: Log rank test.

	* RFS (*n* = 71)				^#^ CSS (*n* = 72)			
Variables	n (Median RFS)	HR	95% CI	*p*-Value	n (Median CSS)	HR	95% CI	*p*-Value
Age (yr)		0.8389	0.2156–0.9088	0.0264		0.5559	0.2484–0.9766	0.0423
≤60	16(11.9)				18(21.4)			
>60	55(21.3)				54(35.0)			
Gender		—	—	0.530		—	—	0.240
Female	33 (16.1)				33 (36.6)			
Male	38 (15.3)				39(26.6)			
Pancreas tumor location		—	—	0.8223		—	—	0.7551
Head	56(13.1)				57(28.7)			
Others	15(22.7)				15(419)			
^Δ^ Pathological AJCC Stage 8th ed.		2.2316	1.4382–4.6816	0.0015		3.2258	2.3430–8.1967	0.0001
I–II	42(24.3)				43(42.7)			
III	29(10.9)				29(16.4)			
^l^ AJCC Positive lymph nodes		2.2457	1.1094–3.4819	0.0206		2.1580	1.0707–3.4294	0.0285
N0	14(41.0)				15(43.7)			
N1–2	57(12.7)				57(24.9)			
Tumor size (mm)		—	—	0.8909		—	—	0.8102
0–20	20(14.9)				20(33.8)			
>20	51(15.1)				52(26.6)			
^‡^ Histologic Tumour grade		—	—	0.1598		—	—	0.1080
G2	65 (16.0)				66(28.7)			
G3	6 (49)				6(41.9)			
^¢^ Margin status after resection R		—	—	0.0895		—	—	0.2013
R0	48(15.1)				50(33.1)			
R1	23(14.9)				22(29.2)			
Vascular Invasion		2.0173	1.1204–5.7405	0.0255		2.1227	1.1315–6.9784	0.0260
No	58(17.7)				61(33.5)			
Yes	13(9.5)				11(19.6)			
Perineural Invasion		4.5956	1.5267–5.1440	0.0009		3.7651	1.3060–4.6577	0.0054
No	11(–)				10 (–)			
Yes	60(12.9)				62 (25.3)			
^ø^ CA 19–9		—	—	0.8940		—	—	0.4018
(0–100 U/mL)	37 (13.1)				38(29.9)			
(100–350 U/mL)	19 (16.0)				18(38.5)			
(>350 U/mL)	15(13.1)				16(21.9)			
^¢^ Histotype		0.1385	0.1642–0.8569	0.0200		—	—	0.0966
Ductal adenocarcinoma	67(13.1)				69(28.7)			
Ductal adenocarcinoma + IPMN	4(–)				3(–)			
Adjuvant chemotherapy		—	—	0.1976		—	—	0.2427
No	10(29.5)				7(49.4)			
Yes	57(14.9)				59(29.9)			
Adjuvant Radiotherapy		—	—	0.7986		—	—	0.6187
No	48(18.1)				47(33.8)			
Yes	19(13.1)				9(24.4)			
Progression local and distant spread		—	—	0.4466		—	—	0.5034
Local	14 (18.1)				14(35.5)			
Distant	41 (11.8)				40(22.8)			
Local and Distant	4 (11.6)				4(24,3)			
Chronic pancreatitis		—	—	0.0980		1.7575	1.0417–3.5112	0.0365
No	48(14.9)				49(31.1)			
Yes	23(18.1)				23(23.3)			
Vascular neoplastic emboli		3.2648	1.1825–4.4287	0.0140		3.3647	1.2124–4.5434	0.0113
No	9(–)				10(–)			
Yes	62(13.1)				62(26.6)			
CXCL12		3.3167	3.0193–14.2857	0.0001		2.4845	1.7280–6.9204	0.0024
Low expression	51(24.9)				52(35.0)			
High expression	20(8.4)				20(19.2)			
CXCR4		0.4572	0.1971–0.7051	0.0024		—	—	0.0839
low expression	46(26.0)				49(36.6)			
High expression	25(11.7)				23(21.9)			
CXCR7		—	—	0.0605		1.7173	1.0162–3.3146	0.0441
Low expression	48(17.6)				48(29.9)			
High expression	23(12.0)				24(29.2)			
PD-L1		—	—	0.1718		—	—	0.3989
Low expression	45(14.9)				46(29.2)			
High expression	26(19.8)				26(36.8)			

# From 76 patients 72 were analyzed for cancer specific survival (CSS), 4/76 patients excluded: cause of death listed as other than cancer. * From 76 patients, 71 patients analyzed for recurrence free survival (RFS), 5/76 patients excluded: cause of data missing or lost to follow-up. Δ American Joint Committee on Cancer (AJCC) Stage 8th edition. l AJCC staging subclassifies lymph node (LN) group: N0 (negative LNs); N1 (1–3 positive LNs) and N2 (≥4 positive LNs). ¢ Margin status after resection R, Surgical resection margin R0, tumor cells from margin distance > 1 mm, R1 tumor cells from margin distance ≥ 1 mm. ‡ Histologic grading system based on extent of glandular differentiation: G1 = well differentiated; G2 = moderately differentiated; G3 = poorly differentiated. ø Serum level of CA-19-9 also called cancer antigen-19-9 or sialylated Lewis. ¢ Intrductal papillary mucinous neoplasm (IPMN). —HR, 95% CI, not reported for variable with *p*-value ≥ 0.05.

**Table 5 cells-11-03340-t005:** Multivariate analysis of cancer specific survival and recurrence-free survival: Cox proportional-hazards regression.

		* RFS (*n* =71)			^#^ CSS (*n* = 72)	
Variables	HR	95% CI	*p*-Value	HR	95% CI	*p*-Value
Age (yr)	—	—	—	—	—	—
≤60						
>60						
^Δ^ Pathological AJCC Stage 8th ed.	1.2125	1.0012–1.4757	0.0490	1.5416	1.2779–1.8597	0.00007
I-II						
III						
AJCC Positive lymph nodes	1.9816	0.8912–4.4062	0.0951	—	—	—
N0						
N1-2						
^¢^ Margin status after resection R	—	—	—			
R0						
R1						
Vascular Invasion	1.808	0.9243–3.5366	0.0852	2.6831	1.3190–5.4581	0.0067
No						
Yes						
Perineural Invasion	3.9024	1.3777–11.0535	0.01077	2.5256	0.9001–7.0869	0.0799
No						
Yes						
Histotype	—	—	—	—	—	—
Ductal adenocarcinoma						
Ductal adenocarcinoma+IPMN						
Adjuvant chemotherapy	—	—	—			
No						
Yes						
Chronic pancreatitis	—	—	—	1.7577	0.9731–3.1750	0.06286
No						
Yes						
Vascular neoplastic emboli	—	—	—	—	—	—
No						
Yes						
CXCL12	3.7184	2.0537–6.7325	0.00002	2.3515	1.2784–4.3253	0.0062
Low expression						
High expression						
CXCR4	—	—	—	—	—	—
low expression						
High expression						
CXCR7	—	—	—	—	—	—
Low expression						
High expression						

# From 76 patients 72 were analyzed for cancer specific survival (CSS), 4/76 patients excluded: cause of death listed as other than cancer. * From 76 patients, 71 patients analyzed for recurrence free survival (RFS), 5/76 patients excluded: cause of data missing or lost to follow-up. Δ merican Joint Committee on Cancer (AJCC) Stage 8th edition. AJCC staging subclassifies lymph node (LN) group: N0 (negative LNs); N1 (1–3 positive LNs) and N2 (≥4 positive LNs). ¢ Margin status after resection R, Surgical resection margin R0, tumor cells from margin distance > 1 mm, R1 tumor cells from margin distance ≥ 1 mm. — HR, 95% CI, not reported for variable with *p*-value ≥ 0.05.

## Data Availability

The datasets used and/or analyzed during the current study are available from the corresponding author or 10.5281/zenodo.6385244.

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
