# Peer review of "CXCR4-CXCL12-CXCR7 and PD-1/PD-L1 in Pancreatic Cancer: CXCL12 Predicts Survival of Radically Resected Patients"

_cells, 2022, doi:10.3390/cells11213340_

Round 1
Reviewer 1 Report
This manuscript reports the prognostic role of CXCL12 evaluated in histological specimens from 76 radically resected pancreatic ductal adenocarcinoma patients and in a small validation cohort of fine needle aspiration cytology specimens. The work and the topic are very interesting and the inclusion of a validation cohort is a plus for the paper although with few patients.
Minor points should be addressed for publication of this interesting paper.
The authors should improve the readability of table 4, for instance indicating the columns of univariate/multivariate analyses, reporting hazard ratio for univariate analysis if statistically significant.
Moreover, the authors should specify, in materials and methods section, the criterion chosen to include variables in multivariate models.
Page 11 line 324-328: Have the authors tested CXCR4 and CXCR7 in multivariate model too? If yes, what variables have been included and where are reported the data?
Multivariate analysis has been conducted for R0 subpopulation too?
In table 4 add all symbols and abbreviations in the figure legend of the table.
Page 17, line 320: the p-value reported in the text for CSS (p=0.014) in R0 patient’s subgroup is different from the one reported in the graph in figure 6, please check it.
Add a descriptive title in table 3.
Check the text for few misspellings or typing errors, e.g. the sentence in page 13 line 351-353 (repetition of “useful”), page 2 line 73 (“represents”),
Author Response
"Please see the attachment."

Reviewer 2 Report
The authors analyzed 76 samples from PDAC specimen and correlated CXCR4, CXCL12, CXCR7 and PD-1/PD-L1 expression of these samples with recurrence-free and overall survival. In a multivariate analysis, pathological tumor stage, vascular ans perineural invasion, chronic pancreatitis and CXCL12 staining were prognostic. In addition, 20 FNP samples were stained.
Comments:
- the manuscript requires extensive language revsion
- the authors do not discuss their results with regard to results of the literature - why is chronic pancreatitis prognostic? PD-1 expression has been reported to be prognostic (eg Diana & al. Oncotarget 2016, Karamitopoulou & al. Cancer Immunol Res 2021). How do the authors explain this descripancy?
- The results of PD-1 are not mentioned in the abstract
- how was "R0" defined?
- in the results section, a subgroup of 48 patients (R0 resection) is mentioned. However, 51 patients underwent R0 resection according to patients characteristics.
The analysis of 20 FNP specimen requires better explanation. How was PDAC proven in these samples?
Author Response
"Please see the attachment."

Round 2
Reviewer 1 Report
The authors answered all points. the work is now acceptable for publication